# Clinical Characteristics and Risk Factors for Intra-Abdominal Infection with *Chryseobacterium indologenes* after Orthotopic Liver Transplantation

**DOI:** 10.3390/pathogens11101126

**Published:** 2022-09-29

**Authors:** Yixin Zhang, Xiaoyu Zhao, Su Xu, Ying Li

**Affiliations:** 1Institute of Antibiotics, Huashan Hospital, Fudan University, Shanghai 200040, China; 2Key Laboratory of Clinical Pharmacology of Antibiotics, National Health Commission, Shanghai 200040, China

**Keywords:** *Chryseobacterium indologenes*, intra-abdominal infections, risk factors, orthotopic liver transplantation

## Abstract

The incidence of hospital-acquired infections caused by *Chryseobacterium indologenes* (*C. indologenes*) is increasing. This study investigated the epidemiological and clinical features of *C. indologenes* intra-abdominal infections in patients who underwent orthotopic liver transplantation (OLT). In this retrospective study, 53 consecutive non-replicate clinical isolates of *C. indologenes* were collected and identified from the OLT patients at a tertiary care university hospital in Shanghai in 2017. Genetic relatedness of the isolates was determined by enterobacterial repetitive intergenic consensus polymerase chain reaction DNA fingerprinting. Antimicrobial susceptibility of the isolates was measured using the microdilution broth method. Nosocomial clonal transmission of *C. indologenes* was confirmed by bacterial homology analysis. All *C. indologenes* isolates were resistant to β-lactams, carbapenems, quinolones, and aminoglycosides, and showed susceptibility to trimethoprim–sulfamethoxazole and minocycline. Multivariate risk modelling revealed that ≥2 bed transfers and an operation time of ≥8 h were independent risk factors for *C. indologenes* intra-abdominal infection after OLT. A nomogram was constructed based on the screened risk factors, which showed good concordance and accuracy. Clonal dissemination of *C. indologenes* in OLT patients was demonstrated and several risk factors for intra-abdominal infections were identified. Epidemiological surveillance of this organism and extensive surveillance programs are imperative worldwide.

## 1. Introduction

*Chryseobacterium indologenes* (*C. indologenes*) has emerged as an important multidrug-resistant (MDR) Gram-negative pathogen which is intrinsically resistant to carbapenems, cephalosporins, aminoglycosides, and chloramphenicol [1]. *C. indologenes* is associated with various infections, especially in immunosuppressed hosts or infants [2,3]. Since it can colonize the hospital environment, cases of *C. indologenes* nosocomial infections have increasingly been reported [4].

Nosocomial infections are particularly prevalent in patients following orthotopic liver transplants (OLTs) due to the complexity of the surgery and the use of anti-rejection drugs [5]. The first two months after transplantation are often plagued by bacterial infections, which negatively impact patients and their grafts [6]. Nosocomial pathogens, including Gram-negative and Gram-positive bacteria, are the most common organisms associated with epidemiological exposure [7]. There are differences in the prevalence of specific pathogens between transplant centers. Recently, the emergence of multidrug-resistant bacteria has become of great concern in OLT patients. Several studies have provided instructive data regarding the effects of infections with extended-spectrum beta-lactamase-producing bacteria and carbapenem-resistant Gram-negative bacteria in OLT patients [8,9,10].

It has been reported that *C. indologenes* may cause subcutaneous port-related bacteremia in liver transplant recipients [11]. As such, *C. indologenes* are also a potential cause of indwelling device infections after transplantation. In its most devastating form, *C.indologenes* infection manifests as clonally transmitted serious pneumonia in critically ill populations and eventually leads to a mortality rate of 25% [4]. However, no study yet has investigated the risk factors for *C. indologenes* infection after OLT. Due to intrinsic multidrug resistance, infections with *C. indologenes* make treatment difficult and a huge burden for immunocompromised patients. In this study, we identified clonal dissemination of *C. indologenes* and conducted a retrospective case–control study to identify patients at increased risk of *C. indologenes* intra-abdominal infections among patients after OLT.

## 2. Materials and Methods

### 2.1. Bacterial Identification and Antimicrobial Susceptibility Testing

This was an OLT-patient-focused sub-study of a previously published *C. indologenes*-based prospective observational study from January 2010 to December 2018 at a tertiary care university hospital in Shanghai, China [12]. The microbial cultures were routinely collected according to the clinical requirements of the OLT patients. Species identification was firstly performed and confirmed by the central laboratory using the Vitek-2 compact system (bioMérieux, Lyon, France). The isolates were stored as glycerol stocks at −80 °C. These strains were resuscitated after aerobic incubation at 37 °C on sheep blood agar for 24 h, and 16s rRNA sequence analysis (type strain: DSM 16777T158; GenBank accession no. LN681561) was performed for re-identification of *C. indologenes*. The minimum inhibitory concentrations (MICs) of 16 antimicrobial agents were measured by the microdilution broth method, according to the Clinical and Laboratory Standards Institute 29th edition (CLSI 29th ed.), as described previously [13]. Briefly, a colony suspension equivalent to 0.5 McFarland standard of each resuscitated strain was inoculated into cation-adjusted Mueller–Hinton broth (*BD Biosciences*, Franklin Lakes, NJ, USA). The results were interpreted after 18 h incubation aerobically with different concentrations of antibacterial drugs at 35 °C ± 2 °C.

### 2.2. Homology Analysis

The enterobacterial repetitive intergenic consensus polymerase chain reaction DNA (ERIC-PCR) fingerprinting method was performed as previously described to analyze the homology of *C. indologene*. After resuscitation from glycerol stock, genomic DNA from each *C. indologenes* strain was extracted for templates, and the primers ERIC1 (5′-ATGTAAGCTCCTGGGGATTCAC-3′) and ERIC2 (5′-AAGTAAGTGACTGGGGTGAGCG-3′) were synthesized [14]. PCR was used to amplify genomic DNA, and fingerprints of genomic DNA were generated. Detection of PCR products was carried out by 2% agarose gel electrophoresis. In order to identify the DNA fingerprinting results, BioNumerisc software version 7.6 (Applied-Maths, Sint-Martens-Latem, Belgium) was utilized, and the genetic analysis was carried out using dice coefficient analysis and the unweighted pair-group method. *C. indologene* strains with a similarity exceeding 70% were considered to be clonally related.

### 2.3. Case–Control Study

The medical records of all patients who underwent an orthotopic liver transplant in 2017 were collected to review the clinical information, including demographic characteristics, main diagnosis, underlying disease, preoperative laboratory test data, surgery, traumatic procedures, symptoms associated with infection, prognosis, etc. A single-center, retrospective study was designed to identify the risk factors for *C. indologenes* intra-abdominal infection after OLT and compare the outcomes of patients with or without *C. indologenes* infection after OLT. The inclusion criteria were as follows: 1. underwent OLT during hospitalization; 2. ≥14 years old. The exclusion criteria were as follows: 1. incomplete or missing medical history; 2. hospitalization duration of less than 48 h. There were no deviations from the Declaration of Helsinki (as revised in 2013) in any of the procedures employed in this study involving human participants. The study was approved by the Ethics Committee of Huashan Hospital, Fudan University, China (approval no. KY2021-1018). Individual consent was waived for this retrospective analysis.

### 2.4. Assessment

Intra-abdominal infections were defined as described in the Centers for Disease Control and Prevention’s National Healthcare Safety Network guidelines [15]. Early postoperative intra-abdominal infections were diagnosed by abnormal hemograms, elevated temperatures, and positive liquid cultures. Anemia was defined as a Hb level of <120 g/L for males and of <110 g/L for females older than 14 years. Hyponatremia was defined as serum sodium <134 mmol/L. Hypokalemia was defined as serum potassium <3.5 mmol/L. Hypoalbuminemia was defined as a serum albumin level of <35 g/L. Hypoprealbuminemia was defined as intravascular prealbumin levels <186 mg/L. Calculation of the model for end-stage liver disease (MELD) scoring was performed using the following formula: MELD score = 3.78 × ln [Total bilirubin (mg/dL)] + 11.2 × ln (INR) + 9.57 × ln [Creatinine (mg/dL)] + 6.43 [16].

### 2.5. Statistical Analysis

All statistical analyses were performed using SPSS.25.0 statistical software (SPSS25.0 Inc., Chicago, IL, United States) and R software version 3.6.1 (http://www.R-project.org, accessed on 19 May 2022). Continuous data were expressed as the means ± standard deviations (SDs) and discrete variables as frequencies and percentages. In order to determine the descriptive analysis, two-sample *t*-tests were applied to continuous variables with normal distribution and Mann–Whitney U tests were applied to continuous variables with non-normal distribution. For categorical variables, Pearson chi-square or Fisher’s exact tests were used. In the logistic regression analysis, univariate regression analysis was used to screen the variables associated with *C. indologenes* infection. Significant variables in the univariable model were used in multivariate regression with backward stepwise selection for the construction of a risk factor model, which was visualized with a nomogram. The prediction model was examined for goodness of fit using the Hosmer–Lemeshow test and an internal validation through the bootstrap method using 1000 bootstrap replicates. The optimal threshold was identified using Youden’s Index, according to which the classification model was evaluated. Predictive accuracy was calculated by utilizing a receiver operating characteristic (ROC) curve. Statistical significance was defined as *p* < 0.05.

## 3. Results

### 3.1. Antimicrobial Susceptibility Testing

Table 1 shows the antimicrobial susceptibility profiles of *C. indologenes* isolates from OLT recipients. The most effective antimicrobial agents were minocycline and trimethoprim–sulfamethoxazole, with 100% susceptibility rates. The isolates demonstrated extreme resistance to cephalosporins (cefotaxime and ceftazidime), β-lactamase/β-lactamase inhibitors (cefoperazone–sulbactam), carbapenems (meropenem and imipenem), quinolones (levofloxacin and ciprofloxacin), and aminoglycosides (amikacin). Piperacillin–tazobactam and rifampicin appeared to be potentially effective antibiotics.

### 3.2. Homology Analysis

We determined the genomic diversity of the 53 strains of *C. indologenes* in the isolates collected from the enrolled patients using the ERIC-PCR DNA fingerprinting method. As shown in the dendrogram, these strains harboured a remarkably consistent ERIC-PCR pattern similarity of 71.4% (Figure 1). The ERIC-genotypic profiles generated were grouped into one cluster at a similarity cutoff of 70%, indicating the clonal spread of *C. indologenes* that was responsible for hospital-acquired infections.

### 3.3. Patient Characteristics 

Over the 1-year study period, 146 patients underwent OLT and 15 patients were excluded (Figure 2). Ultimately, 131 patients were included in the final analysis, of whom 53 (67.9%) developed a microbiologically confirmed infection caused by *C. indologenes*. All *C. indologenes* isolates were collected from the ascites of patients who were diagnosed with intra-abdominal infection after OLT. 

The clinical characteristics of the *C. indologenes* intra-abdominal infections are presented in Table 2. Of the 53 patients with *C. indologenes* infection, 46 (86.8%) were men, with a male-to-female ratio of 6.6:1. The median age at liver transplantation was 49.60 ± 1.48 years, and the mean body mass index (BMI) was 23.97 ± 0.55 kg/m^2^. Of the patients who underwent OLT, 35 (66%) had viral hepatitis B, and 22 (41.5%) had hepatocellular carcinoma. The remaining patients underwent OLT for drug-related liver failure and autoimmune hepatitis cirrhosis. The mean preoperative model for the end-stage liver disease (MELD) scores of patients with *C. indologenes* infection was 17.91 ± 1.16. In addition, 9.4% of *C. indologenes-*infected patients had hypertension, and 11.1% had diabetes mellitus. There were no significant differences between the cases and controls (78 patients without *C. indologenes*) regarding male–female ratio, mean age, BMI, main diagnosis, or preoperative MELD score. Therefore, the two groups were comparable.

### 3.4. Risk Factors for C. indologenes Intra-Abdominal Infection after OLT

To identify independent risk factors for *C. indologenes* intra-abdominal infection, we collected clinical data on the following aspects: preoperative blood test reports, surgical procedures, postoperative bed changes, and indwelling drains (Table 2). Preoperative laboratory test results were collected to assess whether the patient was anaemic, had a combination of electrolyte disturbances, or had low albumin levels. Among the *C. indologenes-*infected patients, 34 (64.2%) had anemia, 28 (52.3%) had hypoalbuminemia, 52 (98.1%) had hypoalbuminemia, 16 (30.2%) had hyponatremia, and 14 (26.4%) had hypokalemia. The rates of hypoalbuminemia, hypoalbuminemia, and hypokalemia in the *C. indologenes* infection group were higher than those in the control group (52.3% vs. 33.3%, *p* = 0.026; 98.1% vs. 87.2%, *p* = 0.028; and 26.4% vs. 12.8%, *p* = 0.048, respectively).

Given that *C. indologenes* infections are intra-abdominal infections, surgical factors may also be involved in the increased risk of infection. The results showed that the infection group had longer total operation times. Operation time was ≥8 h in 49.1% (26/53) of the cases and in 30.8% (24/78) (*p* = 0.034) of the controls. In addition, the proportion of patients who had indwelling abdominal drainage tubes for ≥14 days was higher in the infection group than in the non-infection group (60.4% vs. 39.7%, *p* = 0.020, respectively).

As *C. indologenes* colonises in the hospital setting, bed and ward mobilisation may increase the risk of exposure. The results of this study demonstrated that 62.3% (33/53) of *C. indologenes*-infected patients had two or more bed transfers during hospitalisation and 18.9% (10/53) had ward transfers. In the control group, 5.1% (4/78) had one ward transfer (*p* = 0.012) and 35.9% (28/78) had ≥2 bed transfers during hospitalisation (*p* = 0.003).

Multivariate analysis showed that ≥2 bed transfers (OR, 2.408; 95% CI, 1.112-5.302; *p* = 0.027) and an operation time of ≥8 h (OR, 2.307; 95% CI, 1.088-5.302; *p* = 0.032) were independent risk factors for *C. indologenes* intra-abdominal infection after OLT (Table 3).

### 3.5. Nomogram Model of C. indologenes Infection after OLT

For predicting the risk of of *C. indologenes* infection after LT, a nomogram model that included important Cox predictors was developed (Figure 3). A score was assigned to each item in the model (Table 4), and the sum was the total score. The point on the possibility axis corresponds to *C. indologenes* infection risk (Table 5). For example, a patient who had an operation time ≥ 8 h (64 points), ≥ 2 bed transfers (65 points), and hypoalbuminemia (100 points) would have a total score of 229 points. The patient’s rate of *C. indologenes* infection after OLT would be approximately 50–60%.

### 3.6. Nomogram Model Verification

According to the Hosmer–Lemeshow goodness-of-fit test, the chi-square value was 7.5429 and the *p*-value was 0.4793, which suggested that the fit of the model was good. Similarly, the calibration curve (Figure 4) plotted by the bootstrap method showed consistency between the predicted probability and the actual probability, with a mean absolute error of 0.018. The receiver operating characteristic curve (ROC) suggested that the area under the curve (AUC) was 0.7513, and the optimal threshold point selected based on the Jorden index was 0.466. Parameters at the optimal threshold point were calculated (Table 6).

### 3.7. Post-LT Outcomes

There was no statistical difference between patients with and without *C. indologenes* infection regarding the length of hospital stay and mortality rate during hospitalisation. The mean length of intensive care for patients with *C. indologenes* infection was significantly greater than for controls (11.94 ± 1.07 vs. 9.33 ± 0.77, respectively; *p* = 0.048) (Table 2).

## 4. Discussion

*Chryseobacterium spp.* are non-fermenting Gram-negative bacilli found primarily in soil and water [17]. Within the hospital environment, *Chryseobacterium* can survive in appropriately chlorinated water systems and on wet surfaces of medical devices. They are regarded as rare opportunistic pathogens of human infectious diseases which mainly affect hospitalized patients [1]. Previous studies have identified medically invasive manipulation and the use of catheters or medical devices as major risk factors for *C. indologenes* infection. Moreover, infants, the elderly population, immunocompromised individuals, and patients with diabetes mellitus, hypertension, and malignancies are at high risk of *C. indologenes* infections, even without invasive manipulation or catheter placement [18,19,20,21,22]. In our study, nosocomial clonal dissemination of *C. indologenes* was confirmed by bacterial homology analysis among orthotopic liver transplant recipients in a tertiary care hospital in Shanghai.

The antimicrobial susceptibility profiles of *C. indologenes* isolates, particularly those from patients who have undergone OLT, have not been extensively studied. Previous studies have demonstrated that *C. indologenes* is usually resistant to carbapenems and aminoglycosides but susceptible to β-lactams, β-lactam/lactamase inhibitors, tigecycline, fluoroquinolones, and trimethoprim–sulfamethoxazole. A study in Taiwan evaluated the antimicrobial susceptibility of *C. indologenes* isolates obtained between 2004 and 2011 and reported that the most active anti-infective agent was trimethoprim–sulfamethoxazole (87.6% susceptibility). Tigecycline, levofloxacin, ciprofloxacin, and piperacillin–tazobactam showed reasonable activities, with susceptibility rates of 41.8, 34.4, 31.6, and 29.4%, respectively [3]. In this study, *C. indologenes* isolates were found to be susceptible to trimethoprim–sulfamethoxazole and minocycline but to have high resistance rates to β-lactams, carbapenems, quinolones, and aminoglycosides. In addition to carbapenem resistance, aminoglycoside resistance has also been expected, as *C. indologenes* demonstrates intrinsic phenotypic and genotypic resistance to these antimicrobial drugs. Levofloxacin and ciprofloxacin exhibit undesirable antibacterial activities which are not consistent with other regional data. ERIC-typing was used to investigate the clonal diversity. It was found that the clinical *C. indologenes* strains isolated from OLT patients exhibited a similarity of 71.4% and categorized into one cluster. This pattern showed that the clonal group with the most extreme antimicrobial resistance pattern caused a nosocomial outbreak.

OLT patients have relatively poor nutritional status, and the transplantation procedure is a further blow to the patient’s organism and immune system, thus increasing the risk of infection by conditional pathogenic bacteria, such as *C. indologenes*. The incidence of postoperative bacterial infections in OLT patients during hospitalisation varies from 14% to 68%, according to reports from different countries [6]. There is evidence that MELD scores, alcoholic liver disease, protein malnutrition, and gastrointestinal bleeding are strongly related to postoperative infections [23,24]. A case–control study was conducted to understand the risk factors for *C. indologenes* intra-abdominal infection after OLT. Multivariate analysis revealed that bed transfer ≥2 times was an independent risk factor for *C. indologenes* infection. There has been an increase in the number of intrahospital transfers as hospital staff balance the needs of patient with bed availability [25]. Hospital-acquired infections have been linked to intrahospital transfers, since patients exposed to more environments and contacts within the hospital have a higher risk of contracting pathogens from contaminated surfaces, other patients, and healthcare workers [26]. The odds of developing a hospital-acquired infection increase by 9% for elderly patients with each extra intra-hospital transfer [27]. In addition, similar findings have been reported concerning the number of exposures to roommates per day and the associated risk of methicillin-resistant *Staphylococcus aureus* and vancomycin-resistant *Enterococcus* [28]. In order to minimize contagious pathogenic spread in the hospital environment, physicians should avoid unnecessary bed transfers during their clinical work.

Another independent risk factor for abdominal infection with *C. indologenes* after OLT identified in this study was an operation time of ≥8 h. It has been established that prolonged surgery duration is statistically associated with an increase in surgical site infection rates [29]. According to a previous study, the risk of surgical site infection increases by 14–19% for each extra hour of operation time [30]. The longer the operation time, the more complex the surgical procedure, and the more time the patient’s open incisions are exposed to the environment, increasing the risk of bacterial contamination [31]. Only a few factors contribute to the prolonged operative time that can be modified. Infection-prevention measures must be strictly adhered to in patients undergoing longer targeted or unexpected surgeries.

As a means of predicting prognosis, nomograms can be used by estimating clinical events and incorporating significant prognostic factors into statistical prediction models. As part of this study, we constructed a nomogram including indwelling abdominal drainage, operative time, bed transfer, hypokalemia, and hypoprealbuminemia, which were screened through multivariate logistic regression analysis. Individuals with a higher total score on the nomogram had a greater risk of *C. indologenes* intra-abdominal infection after OLT. This model was stable and reliable in predicting infection risk, as demonstrated by its calibration and ROC curves.

Prognostically, *C. indologenes* infections did not increase in-hospital 30-day mortality among OLT patients but they significantly prolonged the duration of intensive care, from 9.33 ± 0.77 to 11.94 ± 1.07 days, resulting in an increased medical burden on healthcare providers, which is a major burden when medical resources are insufficient in China, especially for transplant patients.

The main limitation of this study is its single-center retrospective design with a corresponding risk of bias. Moreover, *C. indologenes* remains a rare pathogen that causes infectious diseases, and the results of our study may not be generalizable to other centres. Consequently, long-term, large-scale, and high-quality studies are imperative to further understand the impact of *C. indologenes* on human beings.

## 5. Conclusions

The present study revealed nosocomial *C. indologenes* clonal transmission in OLT patients and provided some insight into the risk of *C. indologenes* infection which could motivate new studies to aid the understanding of this pathogen.

## Figures and Tables

**Figure 1 pathogens-11-01126-f001:**
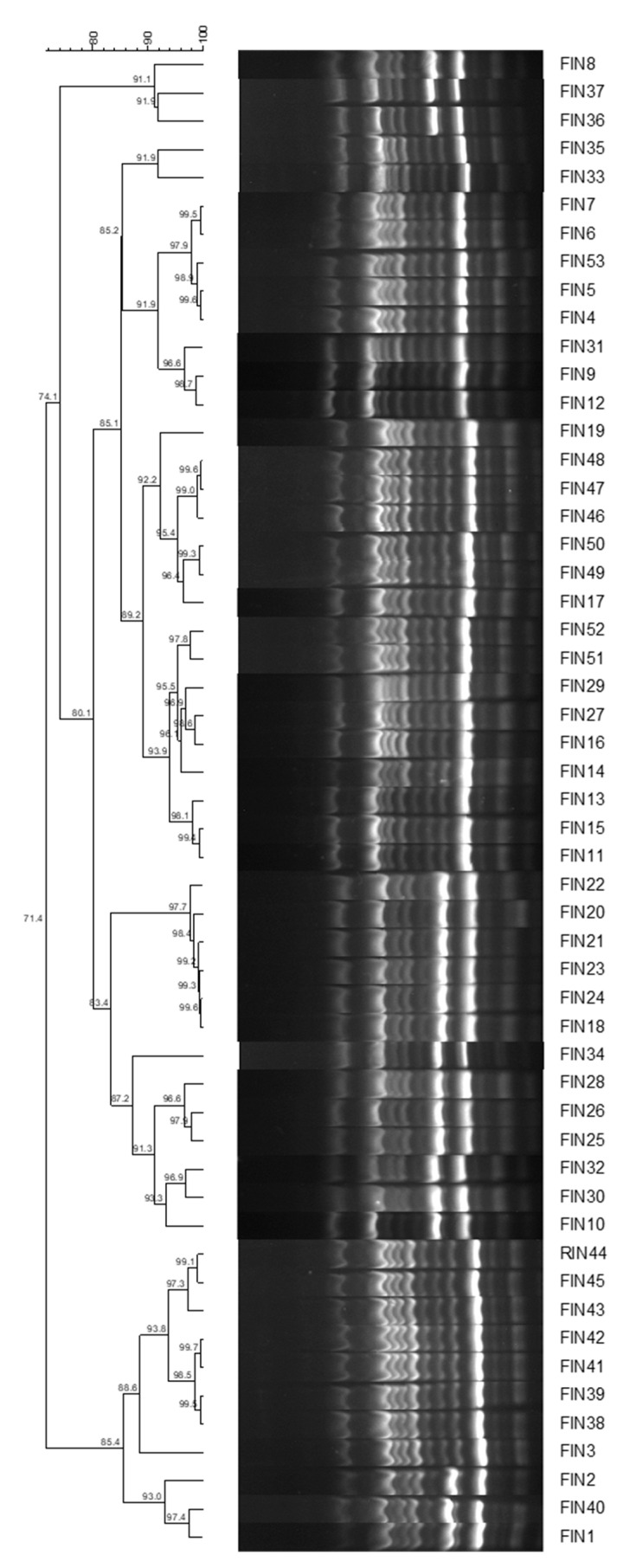
ERIC-PCR-based dendrogram of 53 clinical *Chryseobacterium indologenes* isolates. ERIC-PCR, enterobacterial repetitive intergenic consensus polymerase chain reaction DNA.

**Figure 2 pathogens-11-01126-f002:**
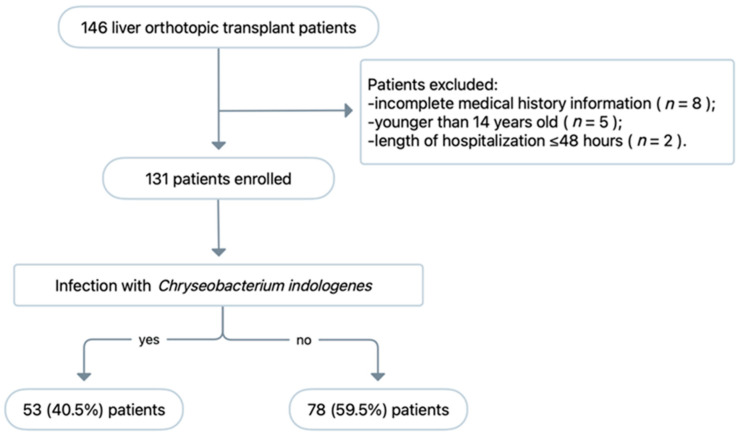
Study flow chart of patient enrolment and exclusion.

**Figure 3 pathogens-11-01126-f003:**
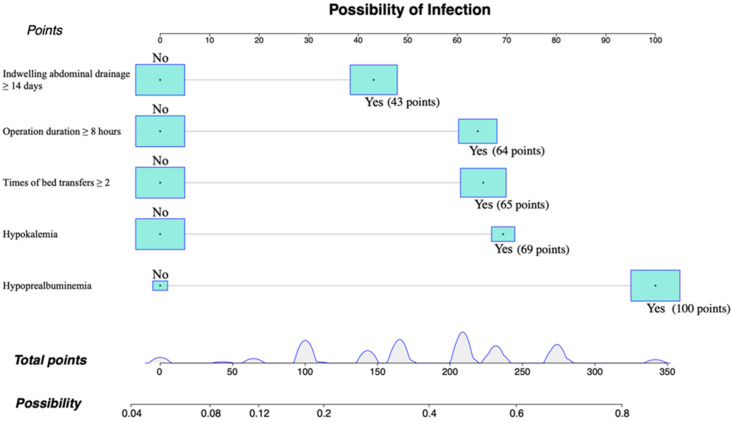
Individualized predictive nomogram model for predicting the risk of *Chryseobacterium indologenes* infection after orthotopic liver transplantation. Version 3.6.1 of R software (R Project for Statistical Computing, Vienna, Austria) was used to create the figure.

**Figure 4 pathogens-11-01126-f004:**
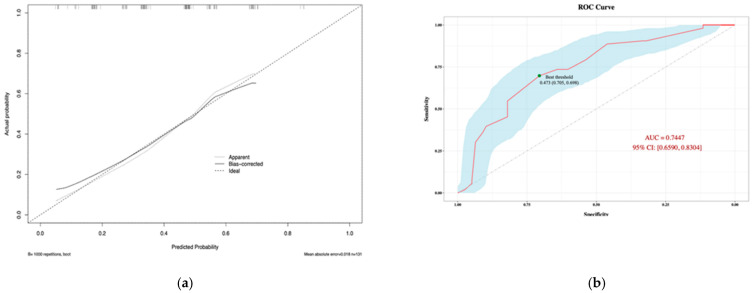
Calibration plot (**a**) and ROC Curve (**b**) of the nomogram model for predicting risk of *Chryseobacterium indologenes* infection after orthotopic liver transplantation. Version 3.6.1 of R software (R Project for Statistical Computing, Vienna, Austria) was used to create the figure.

**Table 1 pathogens-11-01126-t001:** Antimicrobial susceptibility of *Chryseobacterium indologenes* isolated from patients who underwent liver transplantations (μg/mL).

Antimicrobial Agent	S (%)	I (%)	R (%)	MIC_50_	MIC_90_	Susceptibility Breakpoint	MIC Range
Cefotaxime	0.0	0.0	100.0	>128	>128	≤8	128–>128
Ceftazidime	0.0	0.0	100.0	>128	>128	≤8	64–>128
Piperacillin–tazobactam	56.6	7.5	35.8	8	>128/4	≤16/4	2/4–>128/4
Cefoperazone–sulbactam	0.0	0.0	100.0	>128/64	>128/64	≤8/4	32/016–>128/64
Imipenem	0.0	0.0	100.0	>64	>64	≤4	64–>64
Meropenem	0.0	0.0	100.0	>64	>64	≤4	64–>64
Levofloxacin	1.9	0.0	98.1	8	>32	≤2	1–>32
Ciprofloxacin	1.9	0.0	98.1	32	>32	≤1	1–>32
Amikacin	0.0	0.0	100.0	>128	>128	≤16	>128–>128
Minocycline	100.0	0.0	0.0	2	2	≤4	1–4
TMP-SMZ	100.0	0.0	0.0	0.5/9.5	0.5/9.5	≤2/38	<0.125/2.375–1/19
Rifampicin	66.0	1.9	32.1	0.5	16	≤1	<0.25–16

S, susceptible; I, intermediate; R, resistant; MIC, minimal inhibitory concentration; MIC_50_, minimal inhibitory concentration for 50% isolates; MIC_90_, minimal inhibitory concentration for 50% isolates; TMP-SMZ, trimethoprim–sulfamethoxazole.

**Table 2 pathogens-11-01126-t002:** Univariate analysis of clinical characteristics of patients with or without *C. indologenes* infection after liver transplantation.

Characteristics	Infection (*n* = 53)	Without Infection (*n* = 78)	*p*-Value
Male sex/n (%)	46 (86.8%)	72 (92.3%)	0.300
Age			
Range/years	25–75	19–68	/
Mean ± standard deviation/years	49.60 ± 1.48	48.13 ± 1.31	0.463
≥65 years/n (%)	4 (7.5%)	5 (6.4%)	>0.999
BMI/mean ± standard deviation/(kg/m^2^)	23.97 ± 0.55	23.55 ± 0.44	0.548
MELD score	17.91 ± 1.16	16.94 ± 1.03	0.535
Diagnosis/n (%)			
Hepatitis B virus	35 (66.0%)	61 (78.2%)	0.122
Hepatocarcinoma	22 (41.5%)	41 (52.6%)	0.214
Drug-induced liver failure	4 (7.5%)	0 (0)	0.025
Autoimmune hepatitis	3 (5.7%)	1 (1.3%)	0.303
Alcoholic hepatitis	3 (5.7%)	5 (6.4%)	1
Hepatitis C virus	2 (3.8%)	1 (1.3%)	0.565
Congenital liver disease	0 (0)	2 (2.6%)	0.515
Fatty liver	0 (0)	1 (1.3%)	1
Comorbidity/n (%)			
Hypertension	5 (9.4%)	10 (12.8%)	0.550
Diabetes mellitus	6 (11.3%)	10 (12.8%)	0.797
Complications/n (%)			
Anemia	34 (64.2%)	50 (64.1%)	0.995
Hypoalbuminemia	28 (52.3%)	26 (33.3%)	0.026
Hypoprealbuminemia	52 (98.1%)	68 (87.2%)	0.028
Hyponatremia	16 (30.2%)	16 (20.5%)	0.206
Hypokalemia	14 (26.4%)	10 (12.8%)	0.048
Risk factors/n (%)			
Hospital history one month before surgery	30 (56.6%)	39 (50.0%)	0.457
Ward transfer	10 (18.9%)	4 (5.1%)	0.012
Number of bed transfers			
One	18 (34.9%)	37 (47.4%)	0.125
Two or more	33 (62.3%)	28 (35.9%)	0.003
General anesthesia abdominal surgery ≥ 2 times/n (%)	6 (11.3%)	5 (6.4%)	0.351
Operation time ≥ 8 h/n (%)	26 (49.1%)	24 (30.8%)	0.034
Intra-operation blood transfusion volume ≥ 400mL/n (%)	42 (79.2%)	54 (69.2%)	0.204
Tracheotomia/n (%)	3 (5.7%)	3 (3.8%)	0.685
Central venous catheter/n (%)	53 (100.0%)	78 (100.0%)	1
Urethral catheter/n (%)	53 (100.0%)	78 (100.0%)	1
Indwelling urethral catheter ≥ 7 days	21 (39.6%)	33 (42.3%)	0.759
Abdominal drainage catheter/n (%)	46 (86.8%)	60 (76.9%)	0.158
Indwelling abdominal drainage ≥ 14 days/n (%)	32 (60.4%)	31 (39.7%)	0.020
Outcomes			
Intensive care/days ± standard deviation	11.94 ± 1.07	9.33 ± 0.77	0.045
Hospital stay/days ± standard deviation	34.47 ± 2.86	29.58 ± 1.53	0.105
Overall in-hospital mortality/n (%)	4 (7.5%)	9 (11.5%)	0.453

**Table 3 pathogens-11-01126-t003:** Multivariate analysis of clinical characteristics of patients with or without *C. indologenes* infection after liver transplantation.

Risk Factors	Multivariate Analysis		
	OR	95% CI	*p*-Value
Hypoprealbuminemia	3.846	0.644–73.742	0.218
Hypokalemia	2.542	0.976–6.894	0.059
Bed transfer ≥2 times	2.408	1.112–5.312	**0.027**
Operation time ≥8 h	2.373	1.088–5.302	**0.032**
Indwelling abdominal drainage ≥14 days	1.787	0.821–3.934	0.145

Inclusion criterion: α < 0.05; exclusion criterion: α > 0.05. OR, odds ratio; CI, confidence interval.

**Table 4 pathogens-11-01126-t004:** Correspondence between factors and points.

Factor	Points
Indwelling abdominal drainage for ≥14 days	43
Operation duration ≥ 8 h	64
Times of bed transfers ≥ 2	65
Hypokalemia	69
Hypoprealbuminemia	100

**Table 5 pathogens-11-01126-t005:** Correspondence between total points and infection probability.

Total Points	Odds of Infection
<53	<10%
53–113	10–20%
113–153	20–30%
153–186	30–40%
186–216	40–50%
216–246	50–60%
246–279	60–70%
279–319	70–80%
≥320	≥80%

**Table 6 pathogens-11-01126-t006:** Model evaluation at the optimal threshold point according to the Jorden index.

Parameter	Precision	Sensitivity	Specificity	LR+	LR−
Value	0.617	0.698	0.705	2.390	0.418

LR+, positive likelihood ratio; LR−, negative likelihood ratio.

## Data Availability

Not applicable.

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
