# Peer review of "Clinical Characteristics and Risk Factors for Intra-Abdominal Infection with Chryseobacterium indologenes after Orthotopic Liver Transplantation"

_pathogens, 2022, doi:10.3390/pathogens11101126_

Round 1
Reviewer 1 Report
The work is interesting and relevant to medical science. Unfortunately, I have 2 points to improve:
1. In "2.1. Bacterial identification and antimicrobial susceptibility testing", the authors report that C. indologenes isolates were identified by 16S-rRNA sequence analysis. But right away? There were no typical tests done first, such as culture (name of medium, conditions), biochemical identification eg. Vitek, or identification using MALDI-TOF? After all, such research should be done, such as culture. On what medium was the antibiogram obtained?
2. The results in Table 1 are meaningless. For example, the MIC range for Cefotaxime is 1-128 µg/ml. Susceptibility breakpoint is =<8. The MIC range shows that at least 1 strain had MIC = 1 µg/ml, i.e. it was sensitive. However, the percentage for S (sensitive) is 0. Where did this sensitive strain go? Besides, MIC90 is >128? How did it come out when the highest MIC was 128? There were no scores above 128, so it is from the MIC range, but suddenly in the MIC90 they are. The second example is Minocycline, where the MIC range was 0.5-64 ug/ml and the Susceptibility breakpoint was =<4. However, the percentage of S = 100? How is it possible, since at least one result for minocycline is 64, which means that at least one strain should be recognized as R (resistant)? And it actually applies to the entire table, which is totally misinterpreted.
Author Response
Thank you for reading our manuscript and reviewing it, which did help us improve it to a better scientific level.Please see the attachment.

Reviewer 2 Report
I have read with interest the manuscript by Zhang et al. It is a sound paper with valid conclusions.
I have just a few comments to be addressed to improve the manuscript:
C.indologenes infection manifests as clonal-transmitted serious pneumonia in a critically ill population and eventually leads to high morbidity, and a mortality rate of 25%[4]. - please reformulate
Table 1 -maybe there should be pointed out that the resistance percentages are related to the intrinsic resistance of the bacteria, as mentioned in the introduction section.
Section 3.2 - the isolates were stored and you performed the testing, or were all tests performed while the patients were hospitalized? This should be clarified.
row 164 - italicize the bacteria name
row 296 - but significantly prolonged the duration of intensive care - consider adding the word stay.
Please edit the entire bibliography according to the MDPI pattern, as follows:
Journal Articles:
1. Author 1, A.B.; Author 2, C.D. Title of the article. Abbreviated Journal Name Year, Volume, page range.
Author Response

(The authors gave the same response as above.)

Round 2
Reviewer 1 Report
The authors significantly corrected the manuscript according to the reviewer's suggestions. Recently, I recommend the article for publication.